# Parents and Their Children in the Face of Cancer: Parents’ Expectations, Changes in Family Functioning in the Opinion of Caregivers of Children with Neoplastic Diseases—Further Studies

**DOI:** 10.3390/children9101562

**Published:** 2022-10-15

**Authors:** Anna Lewandowska

**Affiliations:** Institute of Healthcare, State School of Technology and Economics, 37-500 Jarosław, Poland; am.lewandowska@poczta.fm; Tel.: +48-698757926

**Keywords:** children, cancer, parents, family, changes

## Abstract

(1) Background: The onset and duration of the child’s illness is a difficult test for the entire family. The stress, in which the family lives from this moment, influences the relationships within it, as well as external relations with the environment. The child’s cancer fundamentally changes the way the family functions, the quality of interactions within the family, and life plans. (2) Patients and Approach: A population survey was run between 2015 and 2020. A cross-sectional study was conducted involving 800 family caregivers of children with cancer during active treatment. The aim of the study was to assess changes in the family functioning in the opinion of parents of a child with cancer. (3) Results: Parents experience chronic anxiety (67%), nervousness (60%), and worry (64%). For 50% of parents, the illness of their child entirely changed their plans. As many as 75% of parents feel guilty for exposing their child to carcinogens. According to the parents, the child’s illness is a psychological (89%), somatic (49%) and financial (77%) burden for them. Only 7% of children cause behavioural problems and 16% have trouble learning due to their sibling’s illness. (4) Conclusions: Cancer is a great difficulty for all members of a family system. The disease disturbs the balance within the family and relations in the family, as well as more distant psychological, social, and material consequences.

## 1. Introduction

Family members of a patient diagnosed with cancer often experience as many difficulties as their ill relative. Although modern medicine is comprehensive, the emotions of the patient and their family will always remain an unresolved problem. Literature and methodology have created many theories, the most important of which are the theories of entangled and separated contributions. More and more studies show that while the symptoms of the patient have an impact on their family, the behaviours of the family members also affect the patient—especially if the patient is a child [1,2,3]. According to Cassileth and Steinfeld, several areas influence family functioning. Firstly, the current model of family interactions may be threatened. Roles are modified, which can bring a sense of loss or overload. Secondly, the illness causes the family to refrain from planning or changing plans due to the feeling of uncertainty. This may cause a sense of suspension or pointlessness. Moreover, the illness not only has an impact on the functioning of the family in external groups but also causes the appearance of new functioning groups [1,4].

Cancer is one of the most serious problems of public health, as chronic diseases are characterised by persistence and constant instability caused by pathological changes, requiring long periods of supervision, absenteeism, care, and rehabilitation. A child’s cancer changes not only their lives, but also influences their environment. The term “collective patient” is used in paediatrics, indicating the fact that the entire family of the ill child is affected by their illness. The family is not only a sum of individuals but a system of interacting elements. To be able to describe it, one needs to know every person being a part of the family, as well as changes occurring within the interactions. The cancer diagnosis in a child, treatment, possible recurrences, experienced side effects, repeated hospitalizations, and visits to the clinic are factors influencing the lives of all family members. They pose a threat to the balance of the entire family system. The patient’s symptoms to some degree influence family members and family members’ behaviour influences the patient. Especially in the case of a child’s cancer, a phenomenon of “family pain” often occurs as a reaction to the illness. The induced stress disturbs the balance of the family system and can cause a crisis. Everything that has so far been stable, valuable, and obvious for the family is at risk. There is a need to reconcile the illness and its treatment with the requirements of everyday life. More and more studies confirming this can be found in the literature [5,6,7].

One of the characteristic traits of a family, in which cancer occurs, is the increased need for cohesion and closeness. Spatial, temporal and worldview barriers among family members disappear. The emotional distance from people outside their family increases. It can be described as a kind of family entanglement, as understood by Minuchin [8,9]. The most important subject is the ill person, the attention of the entire family is focused on them. The boundaries between family members blur, while the external boundaries are often rigid. The most common response to a crisis is family members getting closer to each other. However, other forms of family adaptation to the disease are, e.g., demonstrating hostility and rejection of the sick person. Another characteristic feature of families dealing with cancer is limited communication. Family communication becomes reduced to a simple form and family members avoid confrontation. However, if they can accept the changes that have occurred in the system, the disease can even enrich communication forms in the family. Unfortunately, this does not happen often. Family members are usually left on their own in the face of a crisis. Their entire attention is focused on the ill child and they lose their entire energy in the process of supporting their loved one, which significantly influences the quality of life of the entire family.

The onset and duration of the child’s illness is a difficult test for the entire family. The stress, in which the family lives from this moment, influences the relationships within it, as well as external relations with the environment. The degree of emotional tension and its type, as well as the family’s ability to deal with the problems, change throughout the child’s illness. The stress associated with the child’s chronic disease is greater, the more severe the disease, and the worse the prognosis is. The course of the illness with relapses and periods of remission causes significant mood fluctuations, from fear and anxiety to optimism and joy. The family tries to maintain the balance of the entire system, but to survive, it needs to find ways of adapting to the new situation. Internal organization and the existing rules of functioning change. To cope with the new situation, the family needs to reorganize its functioning. The changes include all areas of the family functioning, including material security, socialization, education and emotional well-being. Overload and excess of duties occur, as well as lack of time for relaxation, lack of emotional contact of the entire family with the ill person, disturbances in performing current professional duties or resignation from work, and lack of time for the remaining members of the family. The child’s cancer fundamentally changes the way the family functions, the quality of interactions within the family and life plans [1,7].

Research on the intensity of emotions occurring during and after treatment has shown that they do not alleviate after therapy. After treatment, these feelings often become more intense due to the fear of possible relapse [10,11,12]. An important aspect is the impact of the child’s illness on their siblings. By spending time together, siblings build lasting relationships that are disturbed by the onset of the disease. Often, emotional disturbances, behavioural changes, and, in the long run, developmental disorders can be observed in siblings of the sick child. Children’s responses to stress vary and may include sleep disorders, somatic symptoms, worse school performance, and even self-destructive behaviours. When parents focus more on their sick child, the rest of the siblings may feel rejected. Their bond with their parents becomes weaker, which often causes a feeling of isolation. Children may avoid talking about their problems and believe that these problems are not currently relevant.

In the literature, more and more studies can be found that confirm the above considerations. However, there are gaps in the subject of the impact of the child’s cancer on the entire family. The family system is built of individual family members who interact while living together. However, it is more than just a sum of personalities, it is a separate whole. Of course, it is most valuable to see the impact of the illness on the family’s life from all perspectives. The study aimed to assess changes in the functioning of the family according to the opinion of caregivers of the sick child and to present the changes from the parent’s perspective.

### 1.1. Methods of Research

The study was descriptive. A technique of non-probability sampling was used. The comparative study was led in the Podkarpackie Oncology Center, Clinical Provincial Hospital in Rzeszów, in 2015–2020. The public health system provided treatment in hospitals and clinics to paediatric cancer patients from Podkarpackie Province. Parents of children diagnosed with cancer were invited to participate in the study in order to assess their opinion about the impact of the illness of their child on the functioning of the family. The participants of this study were included according to the following criteria: being an adult family caregiver of a minor with a confirmed cancer diagnosis and without any chronic or life-threatening illnesses in the past, and the knowledge of the Polish language. All invited caregivers consented to take part in the study. The excluding factor was the diagnosis of cancer given earlier than 3 months ago, as the early stage of treatment can result in a great psychological burden to the patient and their family, which can cause incorrect results. Parents in a bad physical or mental state were also excluded from the study. All invited parents were informed about the research purposes.

### 1.2. Study Participants

800 family caregivers of children in active treatment for cancer took part in a cross-sectional study. To be included in this study, participants must be adults (over 18 years of age), be a parent caregiver of a child with cancer, and provide a signed informed consent form. Illiterate potential participants and parents who did not consent to participate were not included. Only one of the parents, who was the primary caregiver, participated in the study. The primary caregiver was in the hospital during follow-up appointments and hospitalizations, and spent most of the time caring for the child during the child’s illness. Every invited parent received information about the study’s purpose. After they provided their informed consent, parents participated in the interview.

### 1.3. Study Process

The research received the approval of the Bioethics Committee (Resolution No. 386/2009 and 2017/12/4). Respondents took part in the study anonymously, voluntarily, and received information about their right to refuse to participate. They were also aware they could withdraw from the study. Parents received invitations when their child was hospitalized or during a visit. The parents filled in the survey after providing their consent.

### 1.4. Approach

#### Interview Survey

A qualitative, structured, direct, individual, focused, in-depth interview was used to conduct the research. The qualitative interview was a standardized measuring tool, verified by running a test among 30 parents during the month and evaluated for (yielding a Cronbach’s alpha of 0.83). The interview included detailed and extensive instructions for conducting the interview. The first part of the interview concerned parents. It included open-ended, single-choice, and multiple-choice questions that allowed obtaining register and epidemiological information, assessing the general condition of the respondents, as well as their psychological, physical, financial, emotional, everyday life, sexual, spiritual, and care needs. The interview also provided information regarding the occurrence of problems, communication, role, and general functioning. The second part of the interview included information concerning the ill child, provided by the parents: problems, needs, and social functioning. The third part of the interview included information about the siblings of the ill child, provided by the parents: problems, needs, and social functioning. Parents filled in a survey consisting of open-ended, multiple, and single-choice questions, providing records, epidemiological data, and parents’ opinions on the impact of the child’s disease on the functioning of the family.

### 1.5. Process

The researcher, after confirming that the child’s caregiver meets the criteria for inclusion in the study, personally invited them to participate in the study, presented the purpose of the study, and addressed all concerns. In one session, family caregivers provided their informed consent and filled in the survey. There were no consequences for withdrawing consent. Before the filled instruments were collected, they were also checked for any questions left unanswered. The interviewer asked participants to provide answers to all unanswered questions, so no missing values could occur. As the direct interview was conducted, only participants who agreed to participate took part in the study. The number of subjects was imposed by the research period.

The study was in line with the applicable international guidelines of the Helsinki Declaration.

### 1.6. Data Analysis

Data analysis was performed with the SPSS statistical package version 15.0 for Windows. Variables were described as range (minimum and maximum), frequency (%), standard deviation (SD) and mean. The analysis used descriptive statistics and confidence intervals in the assessment of participants’ characteristics, metric and demographic data, and in the assessment of problems. Statistical characteristics of continuous variables were presented in the form of arithmetic means, standard deviations, medians. Statistical characteristics of step and qualitative variables were presented in the form of numerical and percentage distributions, using the Student *t*-test or the Mann–Whitney U test. Correlations were determined using Pearson’s test. Significance was assessed at the level of *p* < 0.05. Kappa Cohen statistics were used to assess the repeatability of answers to individual questions. Missing data were not included in analyses.

## 2. Outcomes

### 2.1. Demography

All participants were legal guardians of children with cancer. The interview was completed by 800 parents. Of 800 people included in the study, 85% were women and 15% were men. The mean age of the mother is 38.2 SD = 7.25, and of the father is 41.1 SD = 7.03. Other descriptive statistics of the studied group are presented in Table 1.

### 2.2. Parents

The research shows that 15% (95% CI: 14–16) of mothers and 24% (95% CI: 22–26) of fathers suffer from chronic illnesses, such as arterial hypertension (41%, 95% CI: 40–43), hypothyroidism (20%, 95% CI: 18–22), asthma (20%, 95% CI: 18–22), cancer (20%, 95% CI: 18–22), and epilepsy (11%, 95% CI: 8–12). Everyone believes that the disease makes it more difficult to take care of a child. Parents experience chronic anxiety (67%, 95% CI: 63–69), nervousness (60%, 95% CI: 58–62), and worry (64%, 95% CI: 61–67). Moreover, some parents have the feeling of inevitable misery (45%, 95% CI: 41–47), difficulties relaxing (42%, 95% CI: 40–46), and often react with anger (35%, 95% CI: 32–37). For 55% (95% CI: 54–56) of parents, each stage of the illness is difficult, for 27% (95% CI: 24–30) the moment of diagnosis and for 18% (95% CI: 14–20) the treatment stage. 54% (95% CI: 52–56) of parents believe that their child’s illness influenced communication within the family. They do not talk about illness, difficult topics or failures. For 50% (95% CI: 48–56) of parents, the illness of their child entirely changed their plans (Figure 1).

In total, 89% (95% CI: 84–90) of parents always accompany their child during the hospital stay and examinations, 9% (95% CI: 6–12) often, and 2% (95% CI: 1–4) rarely. 29% (95% CI: 24–32) consider themselves overprotective parents. 22% (95% CI: 20–25) of parents devote all their free time to their children (Figure 2). Most of the parents participate in the treatment process (67%, 95% CI: 65–70) by talking to doctors, looking for new information about treatment, and reporting concerns.

As many as 75% (95% CI: 74–77) of parents feel guilty for exposing their child to carcinogens: tobacco smoke (35%, 95% CI: 30–37), electromagnetic field (20%, 95% CI: 19–22), polluted air (11%, 95% CI: 10–12), drugs (11%, 95% CI: 10–12), smoked and fried foods (27%, 95% CI: 25–29), low intake of vitamins (23%, 95% CI: 20–25), low intake of calcium (23%, 95% CI: 20–25), high amount of red meat (23%, 95% CI: 20–25), high intake of carbohydrates (23%, 95% CI: 20–25), and low amount of fibre in the child’s diet (15%, 95% CI: 12–17) and high intake of fats (12%, 95% CI: 10–15).

According to the parents, the child’s illness is a psychological (89%, 95% CI: 87–90), somatic (49%, 95% CI: 46–50) and financial (77%, 95% CI: 75–79) burden for them. More than half of the respondents (56%, 95% CI: 54–59) claim that it is a cost-consuming 1/3 of one parent’s salary, and 38% 2/3 of their salary.

Parents’ expectations of medical staff towards their child are mainly safe administration of drugs (69%, 95% CI: 67–70), recognizing and satisfying the child’s needs (66%, 95% CI: 65–67) and reducing pain (58%, 95% CI: 55–60) (Figure 3). Parents’ expectations of themselves are financial support (75%, 95% CI: 74–76) and psychological support (60%, 95% CI: 59–62). Parents are concerned about the following: death of the child (98%, 95% CI: 96–99), recurrence of the disease (41%, 95% CI: 40–43), and financial difficulties (20%, 95% CI: 19–22).

### 2.3. Siblings

55% of the surveyed families have other offspring besides the sick child. Most of them are girls (68%), and the remaining 32% were boys. The mean age of siblings is 10.2 SD = 9.25. 88% (95% CI: 84–90) of children do not have chronic illnesses, however, 4% (95% CI: 2–6) of them have epilepsy, 2% (95% CI: 1–4) celiac disease, 2% (95% CI: 1–4) allergy, 2% (95% CI: 1–4) diabetes, and 2% (95% CI: 1–4) chromosomal syndrome. According to parents, 71% (95% CI: 70–72) of children experience fear and a sense of threat, while 53% (95% CI: 51–55) of children feel depressed (Figure 4). According to parents, 100% of children are supported by their families, 42% (95% CI: 41–44) of children receive support from their friends, and 21% (95% CI: 20–22) receive help from a psychologist. The specialist assessment shows that 5% (95% CI: 4–7) of children have depression. According to parents, 67% (95% CI: 64–70) of children are happy, while 33% (95% CI: 31–35) are unhappy, sad, or depressed. Parents believe that 22% (95% CI: 21–25) of children rarely think about the future and 7% (95% CI: 4–9) never mention this subject. As many as 57% (95% CI: 54–60) of parents have never talked in detail about the health of a child suffering from cancer to the child’s siblings, and 23% (95% CI: 21–25) talked fragmentarily.

Only 7% (95% CI: 4–9) of children cause behavioural problems, and 16% (95% CI: 14–18) have trouble learning due to their siblings’ illness. 74% (95% CI: 73–76) of children are happy to see their peers, but 37% (95% CI: 35–39) of children see their peers sporadically. 45% (95% CI: 44–46) of children engage in various activities and develop their interests, 36% (95% CI: 34–37) engage in activities only after parental encouragement, while the remaining 19% (95% CI: 16–20) are not interested in additional activities. 85% (95% CI: 84–86) of parents believe that their child’s need to play is satisfied, as is the need for activity (86%, 95% CI: 85–87).

The assessment of the impact of the illness on healthy siblings shows that according to parents 34% (95% CI: 33–36) of siblings started to show compassion, care, and love towards their sick brother or sister as well as the entire family, and 21% (95% CI: 20–22) became more mature and reasonable, 31% (95% CI: 30–32) were able to take care of themselves, while 17% (95% CI: 15–19) started helping with everyday family activities.

## 3. Discussion

Numerous authors have shown in their research that having a child with cancer may cause emotional instability, insecurity, and tension among family members. It is described by parents as the most burdensome and overwhelming experience in their lives, which affects their everyday routine [14,15,16]. In the surveyed group of parents, as many as 45% of them have a feeling of inevitable misery, and 35% often react with anger. The challenges include dealing with their own emotions, but also the emotional response of their child to the disease [14]. As shown by Svavarsdottir’s research, the most burdensome activities that parents mentioned are dealing with behavioural problems of the ill child and their siblings, planning and coordinating activities of the family, and supporting a spouse or partner. Parents also need to care for the wellbeing of all their children, while the ill child still needs upbringing and education, despite being in an unusual situation. Therefore, parents have to learn to negotiate between parenting and dealing with the disease [8,14].

Family members experience various emotional states according to the phase of the illness. One of the most serious and recurring states that accompany the illness at every stage is stress. Diagnosis, treatment, remissions, and relapses can cause severe stress that requires psychological and social adjustment for the patient and the family. It affects relations within the family, as well as external relations with the environment. The type and degree of emotional tension and the ability to cope with problems change throughout the child’s illness. Stress disturbs the balance of the family system and may lead to a crisis [1]. Our research shows that for 55% of parents, every stage of the illness is difficult. Chronically, parents experience anxiety (67%), nervousness (60%) and worry (64%). Other authors report anger as the most common first reaction. It is usually directed at medical personnel and the external environment and may be rooted in feelings of guilt, injustice, or misunderstanding. Guilt is also a common feeling. Family members may feel guilty about not being attentive enough and perceive the illness as a punishment for sins [1,17]. This is confirmed by our research, where as many as 75% of parents feel guilty for exposing their child to carcinogens.

The child’s illness is accompanied by a change. The family needs to find its own ways of adapting to the new situation. The internal organisation and the existing rules of functioning change. The change covers all areas of family functioning, from material security, socialisation and upbringing, to the emotional sphere, which is confirmed by our research—according to parents, the child’s illness is an emotional (89%), somatic (49%) and financial (77%) burden. According to other sources, the difficulties include overload and excess duties, limited rest, lack of emotional contact of the family with the ill person, disturbances in the performance of current professional duties or resignation from work, and lack of time for other family members. Cancer changes the way the family functions, the quality of mutual interactions and life plans [17,18,19]. It changes family roles and household chores. The change of roles requires changes in the ways of communication, rebuilding a new relationship, a different system of bonds between members and mutual understanding. Otherwise, conflict situations arise [19,20,21]. Our research also shows that in 50% of parents, their child’s illness entirely changed their plans, and in 54% the illness negatively influenced communication. Moreover, other authors have shown that the parents’ social life is also transformed. Very often, they limit the scope and frequency of their social contacts. They meet their friends and acquaintances less often. Expenditures on culture are limited as well, parents go to the cinema, theatre and concerts less often [17,22,23,24,25,26,27]. Research by Rossi Ferrario et al. shows that 60% of parents have to give up their hobbies and meeting with friends [21]. A sick child also requires modification of the professional life of their parents, which ultimately results in lowering the family’s standard of living and financial sacrifices at the expense of some household members for the benefit of the rest [18,19]. According to research by Ferrario et al., 20% of caregivers have practical or economic difficulties in terms of fulfilling additional care responsibilities [1,21]. In our research, 56% of parents claim that their child’s disease takes 1/3 of one parent’s salary and 38% 2/3 of the salary.

Cancer causes difficulties to the entire family; it also poses a threat to both relationship quality and psychological outcomes. Adaptation to the disease leads to changes in relations and behaviour between spouses, children, and parents [1]. The child’s illness makes parents overprotective of them. As a result, they try to maintain strong, almost symbiotic ties with the child as long as possible and underestimate the need for the child to become independent. At the same time, parents try to compensate the child for these limitations by spoiling them excessively [1]. Chronic disease can also trigger incorrect parent attitudes, such as being overprotective, and avoiding or rejecting the child. Parental attitudes of avoidance or rejection occur mainly in families with disturbed emotional ties, affected by pathology. Rarely does the child’s illness trigger such attitudes in parents who love their children and create valuable educational resources for them. Our research shows that 29% of parents consider themselves overprotective, while 22% devote all their free time to their child.

Stress overload, negative emotions, and feelings of burden can cause individual costs related to the child’s disease, i.e., psychical and mental health impairment in family members [1,22,23,24]. In studies by Chang et al., caregivers reported changed appetite, high blood pressure, pain, fatigue, headaches, and insomnia [25]. Moreover, other studies have shown that chronic stress has a deteriorating effect on the immune system [20,21,22]. In their research, Kielcolt-Glasser et al. report that parents may also experience heart disease, hypertension, and impaired immune function, which makes them more susceptible to various infections and cancers [24]. In our research, 5% of mothers and 4% of fathers reported chronic diseases which, in their opinion, made it difficult to care for a child.

A cancer diagnosis is an important experience and change also for healthy siblings. Recent studies have shown that changes include their mental state and family and social relationships [27]. The severity and quality of emerging emotional, cognitive and behavioural problems depend on many factors. The most important of them include developmental age, the level of biopsychosocial development of a given child, the relationship between them and the patient before falling ill, relationships with parents, and the stage of the sister’s or brother’s illness. Parents focusing on the sick child and devoting all their time to them can cause negative reactions in the child’s siblings, such as anger, jealousy, and even aggression towards the patient and parents [15,16,28,29,30,31,32]. Nolbris et al. indicated that siblings can feel sadness and show anxiety due to the loss of a normal life of sick siblings, as well as regret due to the loss of importance in the family [30]. Wilkins and Woodgate’s qualitative systematic review of childhood experiences of cancer from a sibling perspective shows that siblings feel a sense of loss in terms of daily activities and routine, as well as a loss of family intimacy [31]. Prchal and Landolt demonstrated that siblings had to deal with difficult aspects of cancer, such as their parents’ absence, coping with the patient’s suffering and worse school performance [33]. Other studies also show similar results [31,32]. This is also confirmed by our research, where, according to parents, as many as 71% of children feel fear and a sense of threat, 53% of children feel depressed, 7% of children cause educational problems, and for 16%, siblings’ disease has a negative impact on the child’s learning. According to other sources, the dominant feelings in siblings include worrying about a sick brother or sister, increased competition, increased anger and frustration, a sense of rejection and guilt, as well as a sense of hatred, jealousy, and isolation. Very often, it is the overprotectiveness of parents that adversely affects the functioning of healthy siblings [30,33,34,35,36,37,38]. What is worth emphasising, according to the results of various empirical studies, siblings’ family life is disturbed [29,30,31]. Björk et al. showed that parents devoted all their energy to a child in oncological treatment, which caused the separation of family members and seeking help from others in caring for healthy children [32,39,40,41,42]. In addition to these negative behaviours, there are also positive aspects. Some studies have shown that the experience of cancer can promote the psychological and social development of siblings, including greater responsibility, greater independence and personal growth, positive self-esteem, better social competence, and greater care and compassion for the sick child [35,36,43,44,45]. Nolbris et al. found that in families, in which one of the children had cancer, the bond between siblings became stronger and closer [40]. As confirmed by other researchers, siblings believe that the illness brought family members closer and increased their unity and closeness [30,36,38]. In our research, according to parents, 34% of siblings began to show compassion, care, and love towards their ill brother or sister, and 21% became more mature and reasonable. Perhaps incorrect communication makes the experiences of the youngest family members more difficult. Chui showed that parents may hold back from initiating conversations about cancer with other children due to concerns about the children’s young age and their limited understanding of cancer [32]. According to other studies, siblings hear about or witness their brother or sister’s painful experiences and, due to the lack of knowledge, can develop negative emotions, e.g., anxiety and fear [28,30,31].

This study focuses on the unique needs of caregivers of children with cancer. The obtained data may be valuable for clinical practitioners, as it allows specialists to meet the needs of parents and provide them with proper help and care; it may also be used to improve the standards of working with parents. Data that show that the therapy stage and time since diagnosis are important variables in the process of adaptation seem to be particularly valuable for specialists. The study was limited by the limited size of the sample, time of evaluation, and a small number of items in the interview, which would allow a more in-depth assessment of changes in the functioning of the family. In future studies, the author will most probably deepen the information on the feeling of guilt among parents related to their child’s illness, which was only indicated in these studies. Subsequent studies should focus more on siblings. Another research project will be dedicated to a separate group, that is siblings of an oncologically ill patient, in order to learn about their problems and needs from their perspective.

## 4. Conclusions

Cancer is a challenging experience, both for the family system and its members. The disease affects the balance of the family structure, relations in the family, as well as more distant psychological, social and material consequences. Individual costs for family members may include threats to their physical and mental health from prolonged stress.The reactions of siblings of a child with cancer are very complex, causing changes in relationships and interactions between family members, but also in interpersonal relationships. The emotions and behaviours that arise have an impact on the possibility of adaptation, personal development, developing interdependence with other family members, maintaining individuality, maintaining normality, as well as academic performance.

## Figures and Tables

**Figure 1 children-09-01562-f001:**
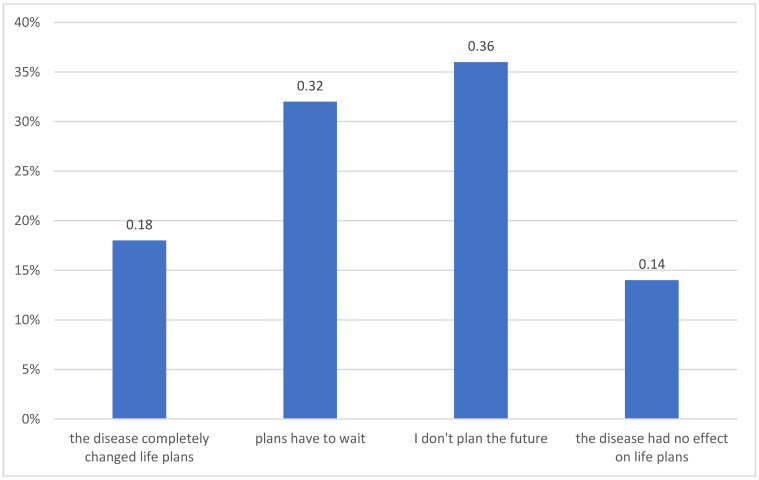
Influence of a child’s neoplastic disease on parents.

**Figure 2 children-09-01562-f002:**
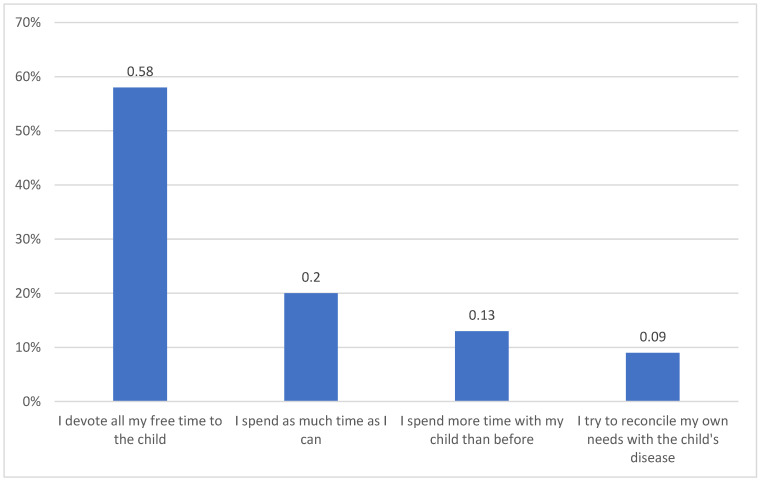
Parental involvement in the child’s disease.

**Figure 3 children-09-01562-f003:**
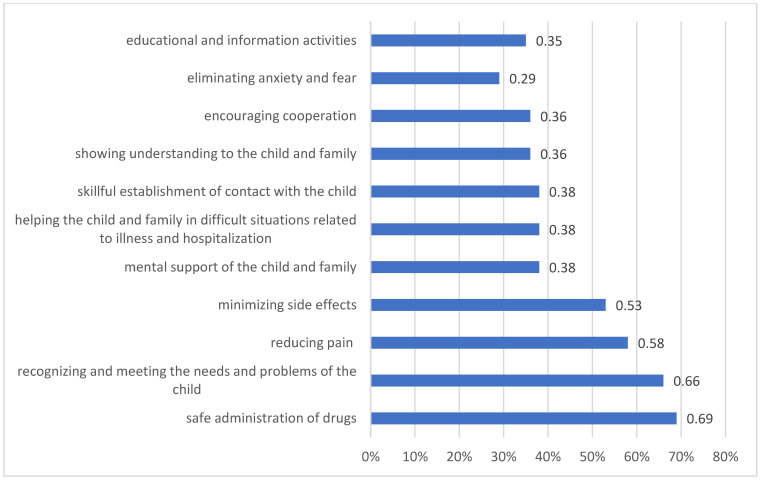
Parents’ expectations towards medical staff.

**Figure 4 children-09-01562-f004:**
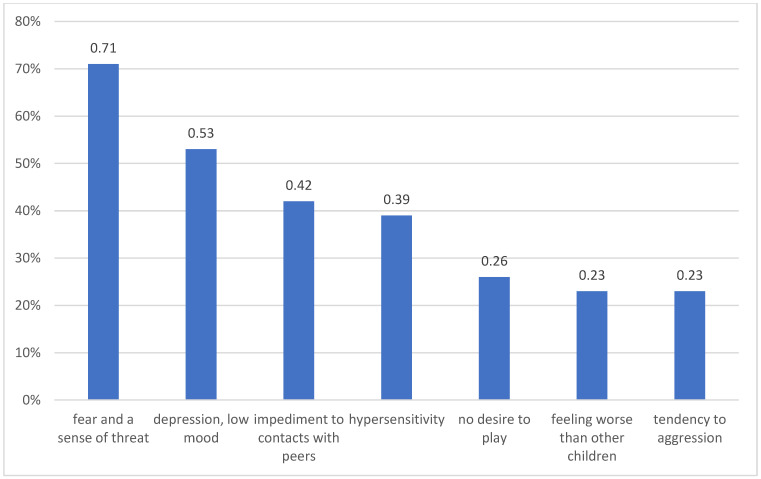
The feelings of siblings of a sick child in the opinion of parents.

**Table 1 children-09-01562-t001:** Descriptive statistics of the examined group [13].

Demographic Information	Total N = 800
Characteristics % (N)	
Sex	
women	85% (680)
men	15% (120)
The age of the study group	
SD	44.1 (7.76)
95% CI	<26; 57>
The age of women	
±standard deviation	38.2 ± 7.25
scope	[26; 57]
median	38
95% CI	[39.8; 41.8]
The age of men	
±standard deviation	41,1 ± 7,03
scope	[26; 57]
median	41
95% CI	[39.8; 41.8]
Place of residence	
city	68% (544)
village	32% (256)
Financial situation	
very good	1% (8)
good	8% (64)
average	68% (544)
bad	10% (80)
very bad	13% (104)
Age groups	
20–29	3% (24)
30–40	35% (280)
41–50	37% (296)
51–60	25% (200)
Education of the study group	
higher education	47% (378)
secondary education	35% (276)
vocational education	18% (146)
primary education	0% (0)
Marital status	
married	74% (592)
widowed	3% (24)
unmarried	23% (184)
Source of income	
professionally active	76% (608)
annuity	15% (120)
benefit	9% (72)
Type of cancer in the family	
leukaemia	54% (432)
brain tumours	19% (152)
solid tumours	27% (216)
Age of children with cancer	
up to 5 years	22% (176)
5–10 years	51% (408)
11–18 years	27% (216)
Number of children owned	
one child	45% (360)
two children	41% (328)
three children	10% (80)
four children	4% (32)
Times of illness	
3–12 months	43% (344)
1–2 years	37% (296)
3–4 years	20% (160)

## Data Availability

Data available on request due to restrictions of privacy and ethics.

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
