# Peer review of "Parents and Their Children in the Face of Cancer: Parents’ Expectations, Changes in Family Functioning in the Opinion of Caregivers of Children with Neoplastic Diseases—Further Studies"

_children, 2022, doi:10.3390/children9101562_

Round 1

Reviewer 1 Report

This study is looking at the important topic of the impact of a child’s cancer diagnosis on the whole family, parents and siblings. It only gathers this information from the parents. The sample size is substantial and the sample seems to be representative of the population. There are however some substantial issues that need addressing. The Introduction is very brief and does not clearly identify the research gap that the study is addressing. A considerable amount of literature mentioned in the Discussion would be better placed in the introduction. The research aims are vague. The study design is unclear, and the questionnaire structure and development is not adequately described. The analyses seem inappropriate and do not adequately explore the dataset. Nor is it clear how they are answering the research question. Analyses are mentioned in the Methods which are not completed in the Results section. The Discussion is detailed and introduces many interesting concepts, however as stated above it would have been better if these were used to justify the study aims and questionnaire design. There also seem to be statements drawn from this study which are not supported by results, but they could be referring to other research, this is not clear. The dataset is substantial and seems to have asked interesting questions, however the analysis completed and the framing of the research question do not seem to adequately interrogate the data.

Abstract

It would be good to include the sample size in the abstract. If there is space it would also be good to include some information on how parents were recruited.

The comment about children in line 20 appears to refer to the sibling without cancer. Please make this clearer.

Introduction

No reference is provided for “Allmond” and the first paragraph has very few references to support the statements. It would be better in the introduction if references were provided for key statements and sentences, not just at the paragraph level. It is difficult to tell if the information provided comes from a single model or theoretical framework. If so, it would be good to explain this.

In preparing to present the objective it should be clearer what the gap is in the literature the author is addressing. The introductory paragraphs do not make clear this gap.

Methods

This section is not very clear and information is repeated in many places. The heading structure is also non-standard and unclear.

Were all eligible parents invited to participate? How many people was this?

The sentence: “Due to the small size of the sample, the share of respondents with fairly consistent characteristics was important was important.” is unclear. Was the population small, or the number of people who consented to take part? What is meant by ‘fairly consistent characteristics’?

Unless all people invited took part, then presumably only those who agreed to participate gave consent? (Line 78)

Were both parents invited where applicable?

The Participants section has repeated information from the study design section. Please provide information in one place only.

Rather than say “refused to volunteer” perhaps “declined to participate” is more sympathetic language to use.

How was the decision about which parent took part made?

How were participants invited to the study? In person? Email? Letter?

Again, there is repeated information in the Research procedures section – it would be better if these sections were more concise and less repetitive.

It is unclear whether participants did a survey or an in depth interview? Did they do both? Lines 94 and 105. Also – see line 95 which refers to an “online questionnaire or a paper version” and line 115 which refers to the interviews lasting an hour.

What is meant by the terms “direct” and “individual” and “focused”.

Please provide more details about the questionnaire such as the number of items in each section, the topics covered, how internal consistency was assessed in a qualitative questionnaire.

Data analysis

Without information on the survey it is difficult to interpret this section. It is not clear what type of information and data has been collected.

Why were comparisons of demographic variables made? How is this answering the research question? No hypothesis was mentioned in the aims so it is not clear what is being measured here.

It is not clear what research questions are being answered by all the analysis mentioned. It would be clearer to state – to answer the …. research question, this.. analysis was conducted.

Results

When presenting demographics such as the percentage of females it does not make sense to present confidence intervals. These are used for inferential statistics.

It is not clear why p values have been calculated when describing the population – what question is being answered? The statistics presented here seem unnecessary.

Much of the data about the parents and siblings would be easier to interpret if it was presented in tables. Again, it is not clear why confidence intervals are being presented.

What are the ages of the siblings?

With no clear information about the structure of the survey/questionnaire and the response options it is difficult to interpret the results.

What is the time since diagnosis for the families involved in the study?

No correlations are presented in the results despite these being mentioned in the analysis section.

Discussion

It would be good to begin the Discussion with a summary of the main findings and how these relate to the main study aim.

Line 227 – references are needed to support the theories mentioned, and more detail should be provided about these.

Line 229 – it is not recommended to use the word “prove” in a scientific paper. A term such as “support” is better.

It would be better to have presented some of the theories about family functioning in the Introduction section. Much of the background literature mentioned in the discussion would be better placed in the introduction, building the case for the design and research questions of the current study.

Statements made in the Discussion need to supported by relevant references.

Line 261 – it is not possible to state that “at every stage of the illness is difficult” as this was not analysed in the results.

Line 355 – what evidence is there for younger siblings being more difficult?

Line 367 – there does not appear to be any data showing that the stage of child’s treatment or the time since diagnosis are important – is this from the literature or from this study?

This study had a large sample size – this would not be a limitation

It is not clear what “evaluation time” refers to in line 369.

Minor

Abstract – line 13 – there is part of a sentence at the end of the line.

Methods – line 75 – I would move “over 18” to straight after “caregiver” otherwise it sounds like the child is over 18 years not the caregiver.

Sentences should not begin with a number presented as numerals.

Table 1 – remove the word ‘owned” after “number of children”

Author Response

Dear Reviewer,

I would like to thank the reviewers for their comments. After analysing all the comments, I made the following changes:

Abstract

The sample size and method of recruiting parents have been added. Fixed a typo error.

Introduction

Literature was added, more evidence was added, and the purpose of the paper was clarified.

Methods

Inaccuracies were clarified, the repetitions were deleted and what was not used in this work was removed from the analysis.

Results

Irregularities have been removed.

Since this work follows on from work previously published in Children, no duplicate tables have been made. On the other hand, a manuscript concerning siblings is being prepared, therefore this article is limited to figures.

Discussion

Meditations on limitations have been expanded.

I hope that the changes made are satisfactory and this will allow publication. I am asking you to take into account the positive comments of the reviewers that this is an interesting study and a good study.

Sincerely

Anna Lewandowska

Reviewer 2 Report

Research on the impact of pediatric cancer on the family system is a very important topic of research.  While I find the study to be interesting, there are several aspects that I would recommend to be addressed.

1. Introduction: The introduction provides a brief overview of the impact of cancer on the family system. Given the results highlighting the data on siblings, I encourage more discussion about the impact on siblings.  

2. Results: While there is likely a limitation to this point, I am curious about where the participants were in their child's cancer treatment when they responded to the questionnaire.  Does time of illness mean total time of treatment for their child or does that mean the amount of time the child has been ill?  

3. Results: Typo in Table 1. - under heading Number of children owned - should be "two" not "dwo"

4. Results: Figure 1 and Figure 2 are the same figure in the manuscript. I believe Figure 2 needs to be uploaded.

5. Results: This recommendation may be limited based on data collected but more demographic information on siblings would be beneficial.

6. Discussion: I would encourage expansion on limitations.  For example, parent report of sibling coping and adjustment is a limitation to this study.  Consider discussion around parent guilt, as the study only examined guilt around exposure to carcinogens.  The construct of parent guilt is much broader than parents feared exposures which are often not cited as causes to childhood cancer.  

7. Abstract: There is an incomplete sentence on line 13..."Patients and"...

Author Response

(The authors gave the same response as above.)
